# Reduction in Allergenicity and Induction of Oral Tolerance of Glycated Tropomyosin from Crab

**DOI:** 10.3390/molecules27062027

**Published:** 2022-03-21

**Authors:** Xin-Yu Han, Tian-Liang Bai, Huang Yang, Yi-Chen Lin, Nai-Ru Ji, Yan-Bo Wang, Ling-Lin Fu, Min-Jie Cao, Jing-Wen Liu, Guang-Ming Liu

**Affiliations:** 1College of Marine Food and Biological Engineering, Fujian Provincial Engineering Technology Research Center of Marine Functional Food, Jimei University, Xiamen 361021, China; 201914908009@jmu.edu.cn (X.-Y.H.); 357900210005@email.ncu.edu.cn (T.-L.B.); 13074803967@163.com (H.Y.); vickycclyc93@gmail.com (Y.-C.L.); 201911832022@jmu.edu.cn (N.-R.J.); mjcao@jmu.edu.cn (M.-J.C.); ljwsbch@163.com (J.-W.L.); 2Food Safety Key Laboratory of Zhejiang Province, School of Food Science and Biotechnology, Zhejiang Gongshang University, Hangzhou 310018, China; wangyb@mail.zjgsu.edu.cn (Y.-B.W.); full1103@hotmail.com (L.-L.F.); 3Zhejiang Engineering Institute of Food Quality and Safety, Zhejiang Gongshang University, Hangzhou 310018, China

**Keywords:** allergenicity, Maillard reaction, mouse models, oral tolerance, *Scylla paramamosain*, tropomyosin

## Abstract

Tropomyosin (TM) is an important crustacean (*Scylla paramamosain*) allergen. This study aimed to assess Maillard-reacted TM (TM-G) induction of allergenic responses with cell and mouse models. We analyzed the difference of sensitization and the ability to induce immune tolerance between TM and TM-G by in vitro and in vivo models, then we compared the relationship between glycation sites of TM-G and epitopes of TM. In the in vitro assay, we discovered that the sensitization of TM-G was lower than TM, and the ability to stimulate mast cell degranulation decreased from 55.07 ± 4.23% to 27.86 ± 3.21%. In the serum of sensitized Balb/c mice, the level of specific IgE produced by TM-G sensitized mice was significantly lower than TM, and the levels of interleukins 4 and interleukins 13 produced by Th2 cells in spleen lymphocytes decreased by 82.35 ± 5.88% and 83.64 ± 9.09%, respectively. In the oral tolerance model, the ratio of Th2/Th1 decreased from 4.05 ± 0.38 to 1.69 ± 0.19. Maillard reaction masked the B cell epitopes of TM and retained some T cell epitopes. Potentially, Maillard reaction products (MRPs) can be used as tolerance inducers for allergen-specific immunotherapy.

## 1. Introduction

Food allergy, a conceptual term which covers many clinical implications, usually refers to an allergic reaction such as a type I hypersensitivity mediated by immunoglobulin E (IgE) [1]. About 8% of infants and 2–3% of adults have a food allergy with increasing incidence year after year [2,3]. Food allergens are the key of food allergy, usually some proteins or proteases with important structural or physiological functions in organisms.

Shellfish is also considered to be one of the most common aquatic products, which can induce severe food allergy diseases, tropomyosin (TM) is the pan-allergen [4]. TM with the 31 to 42 kDa molecular weight (MW), is a type of acidic glycoprotein with good stability to protease, acid, and thermal treatmed [5]. Nowadays, most food products are going through a thermal treatment before consumption, which causes changes in protein structure and physical and chemical properties. Glycation is known as a Maillard reaction (MR) or non-enzymatic browning, which in many studies has been demonstrated to change the allergenicity of food allergens, including Cor a 11, Pru av 1, Mal d 3, Pen a 1 and milk allergen β-lactoglobulin [6,7,8,9,10]. MR of food allergens has been verified as affecting their IgE-binding ability by masking and destroying IgE-binding epitopes. Recently, more and more evidence has shown that Maillard reaction products (MRPs) may affect the interaction of immune cells in the immune system [10,11]. Glycated food allergens may influence the ability of the uptake and presentation by Dendritic cells (DCs), DCs affect the response of downstream immune cells. The food allergens are absorbed by DCs and degraded into peptides by lysosomal after entering the human body. The peptide is recognized by CD4^+^ helper T cells (th) associated with allergy after binding to MHCII molecules [12,13]. In addition, activated Th2 cells secrete interleukins 4 (IL-4) to regulate B cells producing large amounts of IgE [14]. Basophil degranulation has been used to study the allergic reactions with a glycated food allergen, and basophil degranulation was induced by allergen cross-linking of IgE-bound to FcεR I receptors [15].

Oral tolerance is a state of systemic unresponsiveness, which is the default response to food antigens [16]. Failure of immune tolerance induction can cause food allergy [17]. Oral immunotherapy is training for the immune system to achieve a higher threshold, which there might be skewing towards a Th1 response, but mostly oral immunotherapy leads to a regulatory response, which suppresses the Th2 response to the allergen [18]. An ideal tolerance inducer could reduce the production of allergen-specific IgE, activate the tolerogenic DCs, and thus restore Th1/Th2 immune balance to induce regulatory T (Treg) cells responses. In the development of the study on the modulation of the allergic response, the treatment mode using allergen T cell epitopes peptides or processed products instead of allergens as a tolerance inducer was gradually evolved. Gouw et al. [19] demonstrated that the functional human leukocyte antigen DRB1-restricted T cell epitopes were found in the tested hydrolyzed infant formula, which can induce oral tolerance to whey potentially. Wai et al. [20] proved that the T cell epitopes peptides using allergens had the effect of inducing food tolerance after repeated oral administration of allergens. The use of processed food products as tolerance inducers have the characteristics of low production cost, large yield, and low risk [21]. In addition, Liu et al. [22] explored how the TM of crab treated by enzyme cross-linking reaction has the potential to induce oral tolerance in mice, without explaining modifications in the structure of allergenic proteins. Ren et al. [23] revealed the cross-linking reactions of Ara h 2 relevant reaction sites, unfortunately, they found its sensitization potential decreased in the mouse model, no experimental verification of mouse tolerance was carried out.

Antigenic epitopes are divided into two groups based on the cells which can be recognized: T cell epitopes and B cell epitopes [24]. Modifying B-cell epitope of allergens can reduce its allergic reaction, and retaining T-cell epitope of allergens can induce immune tolerance [25,26] Studies have shown that processed food could change allergen allergenicity by destroying antigenic epitopes or creating neoepitopes [27].

There is little data about the MRPs absorption and presentation of DCs, although it is important to explain the allergic responses induced by MRPs. Less information reported that MRPs could be as tolerance inducers. Whether the crab TM treated by MR has the potential to induce oral tolerance requires further exploration. Glycation sites of MRPs should be identified and the relation of epitopes and oral tolerance require further exploration.

In the present study, galactose could effectively decrease allergenicity in TM through MR in vitro assays. Besides, the mechanism of lower allergenicity and induced oral tolerance of Maillard-reacted TM (TM-G) was evaluated with cell and mouse models. Finally, the glycation sites of TM-G were identified to analyze the modification side on the T cell epitopes or B cell epitopes. Overall, these results would form the foundation to contribute to the future development of the Maillard reaction products as a functional food for tolerance induction.

## 2. Materials and Methods

### 2.1. Chemicals

Galactose was purchased from Macklin (Shanghai, China). Rabbit anti-crab TM IgG pAb was prepared in our laboratory previously and goat anti-rabbit IgG-HRP antibody was from Abmart (Berkeley Heights, NJ, USA). Granulocyte macrophage colony stimulating factor (GM-CSF), IL-4 were obtained from Sigma-Aldrich (St. Louis, MO, USA). RPMI 1640 and fetal bovine sera (FBS) were purchased from Invitrogen (New York, NY, USA). CD11c for flow cytometry analysis were purchased from Biolegend (San Diego, CA, USA). Imject Alum was purchased from Thermo Fisher Scientific Inc. (Waltham, MA, USA). Goat anti-mouse IgE, IgG1, and IgG2a antibodies were purchased from Abcam (Cambridge, MA, USA). The ELISA kits of IL-4, IL-10, IL-13, interferon (IFN)-γ, mouse mast-cell protease-1 (mMCP-1) and transforming growth factor-β (TGF-β) were purchased from R&D Systems (Minneapoils, MN, USA). An ELISA kit of histamine was purchased from IBL (Hamburg, Germany).

### 2.2. Animals

Live crab (*S. paramamosain*) was purchased from a local aquatic products market in Xiamen. Specific pathogen-free female Balb/c mice weighing 14–16 g (4–6-weeks-old) were obtained from Shanghai Laboratory Animal Center of Chinese Academy of Sciences (Shanghai, China). The housing environment for mice under specific pathogen-free conditions was maintained at 20–25 °C and 60 ± 10% relative humidity with a 12 h light/dark cycle. Protocols were approved by the Institutional Animal Care and Use Committee of Animal Laboratory Center of Jimei University (Xiamen, China, No. SCXK 2016-0006), and all animals were used for academic research.

### 2.3. Human Sera

Sera were obtained by the Second Affiliated Hospital of Xiamen Medical College (Human ethical approval 2020018, Xiamen, Fujian, China), and signed informed consent was obtained from all individuals. Sera IgE antibodies (Appendix A) to crab were measured by ImmunoCAP (Phadia AB, Uppsala, Sweden), when crab-specific IgE ≥ 0.35 kUA/L were defined as positive, the specific IgE levels with crab TM of positive sera were measured by ELISA previously, negative sera IgE < 0.35 kUA/L, and all sera were stored at −80 °C until further use.

### 2.4. Preparation and Immunobinding Capacity Analysis of TM-G

The purification of TM from *S. paramamosain* and preparing TM-G were accorded as described previously [28]. All samples were dialyzed by 10 mM PBS (pH 7.0) after reaction and then detected by 12% SDS-PAGE, western blot, and dot blot. Immune-binding capacity analysis was performed as described elsewhere with some modifications. Briefly, rabbit anti-crab TM IgG pAb with 1:1 × 10^4^ dilution and HRP-labeled goat anti-rabbit IgG with 1:2 × 10^4^ dilutions were used as the primary and secondary antibody, respectively, for immunoblot analysis. TM and TM-G were adjusted to 1 mg/mL to dot blot, the primary antibody was human sera (dilution 1:3), the secondary antibody was HRP-labeled goat anti-human IgE antibody (dilution 1:1 × 10^4^).

### 2.5. Allergen Uptake and Presentation by Mouse Bone Marrow-Derived Dendritic Cells

Mouse bone marrow-derived dendritic cells (BMDCs) isolated from BALB/c mice were referred to the classical method by Perusko et al. [10]. After being cultured for 6 days, immature BMDCs were collected and used for the following experiments.

TM and TM-G were labeled with FITC according to the method of Liu et al. [23]. Briefly, FITC-proteins (TM, TM-G) were separated from unreacted FITC label using ultrafiltration centrifugal tube (MW cut off 3 kDa). Immature BMDCs were cultured at 1 × 10^5^ cells per well in 24-well plates and incubated gradient time with 50 μg FITC-protein for 0, 10, 20, 30, 40, 50, 60, 90, 120, and 180 min at 37 °C. Then, the collected cells were stained with APC anti-mouse CD11c for 30 min at 4 °C for BMDCs uptake analysis by flow cytometry.

Similarly, immature BMDCs were cultured at 1 × 10^5^ cells per well in 24-well plates and incubated 48 h with 50 μg proteins (TM, TM-G). The total RNA kit and TIANScript RT Kit (Tiangen, Beijing, China) was used to extract the cDNA from BMDCs for RT-PCR, according to the instructions with 20 μL final volume. The changes of CD86 and H2-Ob transcription levels in BMDCs stimulated by PBS, TM, and TM-G were detected by RT-PCR. The following primers were used for RT-PCR: CD86-F, 5′-ATG TCA CAA GAA GCC GAA TC-3′. CD86-R, 5′-TTC AGT GCT CTT GGC CTA TG-3′. H2-Ob-F, 5′-GTG ACC TGG GGA TGT TTG TTG-3′. H2-Ob-R, 5′-CAG GAG ATC CAG GCG TTT GTT-3′. β-actin-F, 5′-ACA GCA CCG TGT TGG CGT AGA G-3′. β-actin-R, 5′-TGC GGG ACA TCA AGG AGA AGC-3′. The result was chosen 2^−^^ΔΔCt^ method calculation the relative quantitative of CD86 and H2-Ob.

### 2.6. Degranulation Assay in RBL-2H3 Cells

RBL-2H3 was purchased from the American Type Culture Collection (Bethesda, MD, USA) and cultured as described previously [10]. MTT assay and assessment of the β-hexosaminidase release rate were conducted according to the method used by Han et al. [28]. RBL-2H3 cells were inoculated on 96-well plates and cultured at 37 °C for 12 h. The cells were washed three times with PBS, and then different concentrations of TM and TM-G were added to stimulate RBL-2H3 cells for 6 h, and then the MTT was added, after 4 h the value was measured at OD_570_ nm to assess cytotoxicity. Assessment of the β-hexosaminidase release rate, the serum of TM sensitized mice with RBL-2H3 cells was incubated for 16 h. Then allergic reactions were induced by co-incubation with PBS, TM-G, and TM [28].

### 2.7. Animal Models of Injection Sensitization and Oral Tolerance to TM-G

Injection sensitization and oral tolerance tests were based on Han et al. [28] with modifications. Mice were allocated randomly to three groups (*n* = 8). For the sensitization model, one week of adaptive culture female Balb/c mice were, respectively, exposed to TM (150 μg) or TM-G (150 μg) with alum adjuvant (6 mg) by injection on days 0 and 14. Mouse blood and spleen lymph cells were collected on day 15. The levels of TM-specific IgE, IgG1, and IgG2a in sera were measured using ELISA. After 3 days of culture, the cytokine assay of spleen lymph cells was measured by ELISA. For oral tolerance, exposed to TM (150 μg) with alum adjuvant (6 mg) by injection on days 0 and 14; next, on days 21 and 28, mice received 2 mg TM or TM-G via intragastric gavage. On day 33, 10 mg TM was given to the mouse. The diarrhea rate within one hour after three times (days 21, 28, and 33) of gavage was recorded, and the anaphylactic score and rectal temperature of mice were measured one hour after gavage on day 33. The method used for the anaphylactic score was according to Liu et al. [29]. Furthermore, blood, spleen, and mesentery lymph cells were collected on day 34 [22,28,29] The mMCP-1 and TGF-β were determined by ELISA kit using mouse sera. The IL-4, IL-10, IL-13, and IFN-γ were determined by ELISA kit using cellular supernatant of the cultured spleen, and mesentery lymph cells.

### 2.8. Identification of Glycation Sites and Modification Analysis of the Epitopes

The amino acid sequence of crab of TM was obtained from NCBI (GenBank number: ABS12233.1). The predicted T cell epitopes’ results from the Prediction System (http://tools.iedb.org/mhcii/ (accessed on 12 January 2022)) were analyzed based on Gouw et al. [19,30,31]. B cell epitopes of TM were referred to Liu et al. [32]. Glycation sites testing was according to Han et al. [28] with modifications. The band of TM-G was hydrolyzed using trypsin (Promega, Madison, WI, USA) for 20 h, and then separated by capillary high-performance liquid chromatography (Orbitrap-ELite, Thermo, Finnigan, San Jose, CA, USA). The separated products were analyzed by Q-Exactive mass spectrometer. The result was named as “RAW file” and analyzed using the Maxquant software to find the modified amino acids. Finally, the sites that were modified by MR were compared with TM epitope.

### 2.9. Statistical Analysis

Data were presented as means ± standard deviation (SD). Statistical significance was assessed using a two-way ANOVA. **^#^**
*p* < 0.05 and **^##^**
*p* < 0.01 was used to compare with PBS group. * *p* < 0.05 and ** *p* < 0.01 was used to compare with TM group.

## 3. Results

### 3.1. IgG/IgE-Binding Capacity Analysis of TM-G

The bands of TM and TM-G are represented in Figure 1A. The MW of TM-G was higher than that of TM, and the band of TM-G performed diffusion. Moreover, the IgG/IgE-binding capacity of TM-G was lower than that of TM (Figure 1B,C) using western blot and dot blot. The band of TM-G was not bound with the specific IgG antibody of TM. Compared with TM, TM-G exhibited weaker binding capacity with patients’ sera.

### 3.2. TM-G Uptake by BMDCs

Furthermore, to investigate whether glycation affects interactions of TM with BMDCs, TM and TM-G with FITC-label were incubated with BMDCs, and then the uptake ability of TM and TM-G was detected in time by flow cytometry. Uptake of TM and TM-G measured as the percent of CD11c^+^ and FITC-positive cells were time-dependent, while the mean fluorescence intensity continuously increased up to 180 min (Figure 2A), the gradient time were 0, 10, 20, 30, 40, 50, 60, 90, 120, and 180 min. TM-G had lower efficiently, reaching 72.53% of FITC-positive BMDCs, whereas BMDCs were incubated with FITC-labeled TM, the uptake ability was up to 94.23% (Appendix A).

### 3.3. TM-G Presentation by BMDCs

The changes of CD86 and H2-Ob transcription levels in BMDCs cells stimulated by TM and TM-G were detected by RT-PCR. Compared with the PBS group, the expression level of CD86 on BMDCs increased by 37.60 ± 20.12% after stimulation in the TM group. However, the expression level of CD86 on BMDCs stimulated by the TM-G group was significantly lower than that of the TM group (*p* < 0.01), without significant difference to the PBS group (Figure 2B). Furthermore, the expression level of H2-Ob was significantly up-regulated in the TM-G group in comparison to the TM group (*p* < 0.01), compared with the PBS group, the expression level of H2-Ob decreased by 6.24 ± 1.02% after stimulation in the TM group (Figure 2C).

### 3.4. Effect of TM-G on Cell Activity and Degranulation in RBL-2H3

MTT assay was used to assess the cytotoxic effects of TM-G by RBL-2H3 cells. It was found that there was no effect on cell viability after TM and PBS stimulation. The cell viability rate of TM-G was more than 90% at different concentrations without significant difference to TM group (Figure 3A). The sera of TM sensitized mice, using the 15-days injection model, was collected to sensitize RBL-2H3 cells. The effect of TM and TM-G on degranulation of RBL-2H3 cells was shown in Figure 3B, compared with the PBS group, the degranulation rate of the TM group was significantly decreased (*p* < 0.01), and the degranulation rate of the TM-G group was significantly decreased compared with the TM group (*p* < 0.01); meanwhile, the degranulation rate of the TM-G group was significantly increased compared with the PBS group (*p* < 0.05).

### 3.5. Injection Sensitization to TM-G

In vivo experiments further proved our hypothesis. The effect of TM and TM-G on mice can be most intuitively analyzed by using the detection of sera antibodies. A 15-day mouse model was used to test the allergenicity of TM-G in vivo (Figure 4A). On the 15th day, the level of serum IgE in the TM group was significantly higher than the PBS group, which proved that the TM sensitized-mouse model was successfully constructed. In Figure 4B, the specific IgE level in the TM group performed the significant difference in the PBS group (*p* < 0.01), and there was no difference in the PBS group and the TM-G group, whereas the specific IgE level in the TM-G group was only one-tenth of that in the TM group (*p* < 0.01). Additionally, the level of IgG1 and IgG2a in the TM and TM-G group was higher than the PBS group, but the level of IgG1 and IgG2a in the TM-G group was significantly lower than the TM group (*p* < 0.01). Meanwhile, the released level of the Th1-type cytokines IFN-γ in spleen cells was similar to both the PBS group and the TM-G group; however, the IFN-γ value of the TM group was significantly lower than the TM-G group (*p* < 0.01) (Figure 4C). The level of released IL-4 and IL-13, cytokine from the Th2-type cell, is shown in Figure 4D,E, the TM group was significantly increased in comparison to the PBS group (*p* < 0.01), whereas the TM-G group was significantly decreased compared with the TM group (*p* < 0.01). Overall, it is further demonstrated that MR could reduce the immunogenicity of allergens.

### 3.6. Oral Tolerance to TM-G

Furthermore, the potential of TM-G supporting the development of oral tolerance in mice was explored with a mouse model (Figure 5A). Anaphylaxis in mice was assessed by anaphylactic score, rectal temperature, and diarrhea rate (Figure 5B–D). All the figures showed the same result, the TM group can induce a significantly anaphylactic reaction in mice through the higher anaphylactic score, lower rectal temperature, and higher diarrhea rate in comparison to the PBS group, while the group of TM-G can relieve these symptoms.

Further, assessing the tolerance potency of TM-G at sera level, normally the activation of mast cells was evaluated by specific antibody, histamine, and mMCP-1 (Figure 5E–G). Compared to the PBS group, the TM group had a higher level in those indexes, whereas the TM-G group alleviated these symptoms. The T-cell level (Figure 5H–L) showed the same results. As the response of Treg cells, TGF-β and IL-10 in the TM group significantly decreased than in the PBS group, while the TM-G group performed a similar degree with the PBS group (Figure 5H,I). Both IL-13 and IL-4 are generally involved in the response of Th2 cells, the TM group had a higher level than the PBS group, and the TM-G group mitigated this phenomenon, while the IFN-γ response of Th1 cells in the TM-G group increased slightly. The index of Th2/Th1 was regarded by IL-4 to IFN-γ ratio. As shown in Figure 5M, TM-G, but not TM, can regulate the balance of Th2/Th1 cells in mice. The up-regulation of IL-10 and TGF-β suggested that Treg cells inhibit allergic reactions by regulating the balance of Th2/Th1, thus relieving allergic symptoms. These results indicated the oral tolerance capacity of TM-G. The MRPs could promote the balance of Th2 cells and Th1 cells in mouse spleen lymphocytes, making them tend to the cell level of normal mice.

### 3.7. Modification of TM Epitopes via the MR

The results of the prediction of T cell epitopes can support information (Appendix A). The main T cell epitopes are distributed in the M1-D20, S102-M126, E145-R182, E219-Q247. The glycation site of TM-G was identified using LC-MS/MS (Appendix A). Glycated peptides were identified by molecular mass compared with the TM peptide masses, which showed in Appendix A.

To compare the epitopes and modification sites, the glycated residues of TM-G were marked by a solid triangle, the predicted T cell epitopes of TM are marked by single-underline, and reported B cell epitopes of TM are marked by double-underline (Figure 6). It was found that many glycated residues were located at the T cell epitopes or reported B cell epitopes. The modified specific amino acids on the B cell epitopes of the TM were R21, R101, R105, R160, K161, R178, R182, K213, R217, and this observation provides the mechanistic explanation of the decreased IgE capability of TM-G.

## 4. Discussion

MR usually occurrs in processed food to improve food quality and taste, and now MR is gradually used to reduce the sensitivity of food allergens. In the previous study, we found that both galactose and arabinose could reduce the allergenicity of TM, while galactose worked better than arabinose [28]. In this study, the mechanism has been further clarified. We found that the IgG/E-binding activity of TM-G was reduced, Nakamura et al. [9] found that the allergenicity of squid TM after MR was reduced, which is similar to this study, and the changes of the immune binding capacity of MRPs to mouse-specific IgE antibody could be detected indirectly by the RBL-2H3 cell model. TM-G could significantly inhibit the β-hexosaminidase release rate, a similar study found that after the Maillard reaction, the degranulation of KU812 cells also decreased [33], which indicated that the immune binding activity of MRPs to IgE antibody from mouse and human serum were also significantly decreased.

To analyze the immunogenic properties of TM-G, the influences on the uptake activity of DCs by TM-G were investigated. Perusko et al. [10] found that the phagocytic function of DCs to milk allergen after MR was slightly enhanced, which is similar to this study. It had a strong phagocytosis ability toward glycosylation products, due to DCs having a variety of cell surface receptors. CD86 is a co-stimulatory factor on DCs, which can activate T cells, H2-Ob has the function of inhibiting and regulating DCs’ presentation [34]. Rupa et al. [35] found that ovalbumin after MR could inhibit the activation of T cells by DCs. The co-stimulatory factor CD86 on DCs’ surface which could activate T cell surface receptor was decreased [17]. In this study, T cell epitopes were not identified, and they will be explored and elaborated on at a later stage. This result may be due to reduced immunogenicity. The in vivo experiments further supported our hypothesis. The TM-specific IgE antibody in mouse serum was decreased in the mouse sensitized model, which further demonstrated the effect of MR on allergen immunogenicity reduction. In this study, the MRPs did not affect the uptake function of DCs; however MRPs could reduce the presentation level of DCs and the signal transduction effect on downstream CD4^+^ T cells. 

Furthermore, the potential of TM-G to support the development of oral tolerance in mice was explored. It has been previously shown that MRPs could promote the balance of Th2 cells and Th1 cells in mouse spleen lymphocytes, making them reach the cell level of normal mice [28]. Treg cells mediate specific suppression by depleting peptide-MHC-II from DCs [36]. The activation of Treg cells is helpful to induce oral tolerance [37]. Multiple tolerance induction makes the body increase the potential of producing immune tolerance. Furthermore, IgG1^+^ B cells do not compete with IgE^+^ B cells as IgE^+^ B cells cannot switch towards IgG1; moreover, allergen-specific IgG1 might block IgE binding to the allergen [14].

Meanwhile, we explained the molecular mechanism of inducing tolerance of TM-G. Though the molecular weight of amino acids on the peptide can be analyzed by mass spectrometry [38], the amino acid changes in TM-G were obtained. We found that TM-G has more glycosylated residues than TM with Maillard reaction with arabinose [28] and many glycosylated residues are located at T cell epitope or reported B cell epitope. Thus, the reaction of T cells could be activated without reacting with IgE on activating effector cells, so as to reduce allergic symptoms and improve the tolerance of the body.

In this study, the glycosylation for TM could reduce its sensitivity and preserve immunogenicity, aimed at improving the efficacy of TM tolerance induction and reduce TM occurrence of serious side effects. In theory, immunogenicity and a low sensitivity allergen vaccine can be used in high doses regardless of the rapid response mediated by IgE, for the significant decrease in the effect of TM-G on the degranulation of mast cells, but still retains the ability to induce Th1 cells to produce IFN-γ [39]. T cell epitopes of TM-G may be modified by MR, MRPs will affect the presentation function of DCs and then affect the immunogenicity of allergen. For allergen tolerance induction with modified low allergenicity protein vaccine, the dosage of tolerance inducer is critical [39]. Individualized treatment should be made according to the specific allergic protein of the patient [40]. Allergen-specific immunotherapy refers to the stimulation of a certain small dose of allergen in allergic patients after the clinical diagnosis of an allergen, and then gradually increase the dose to achieve the purpose of desensitization [41]. Therefore, the MRPs, which can be used as a vaccine or as inducing food, have a great application prospect, but the safety and effectiveness of the MRPs as therapeutic vaccines and food need to be evaluated in clinical trials.

## 5. Conclusions

In summary, due to specific amino acid residues on the T cell epitopes and the B cell epitopes of TM being glycated, the TM-G could reduce the degranulation release rate, and inhibit the uptake and presentation function of BMDCs. Additionally, the IgE levels in mice sera of the TM-G group were significantly decreased in the mouse sensitized model. The TM-G could activate Treg to produce TGF-β and IL-10 without causing allergic symptoms. TM-G positively modulates the Th1/Th2 immune balance, which can decrease the specific IgE antibody and histamine in sera. Potentially, the MRPs as tolerance inducers are used for allergen-specific immunotherapy.

## Figures and Tables

**Figure 1 molecules-27-02027-f001:**
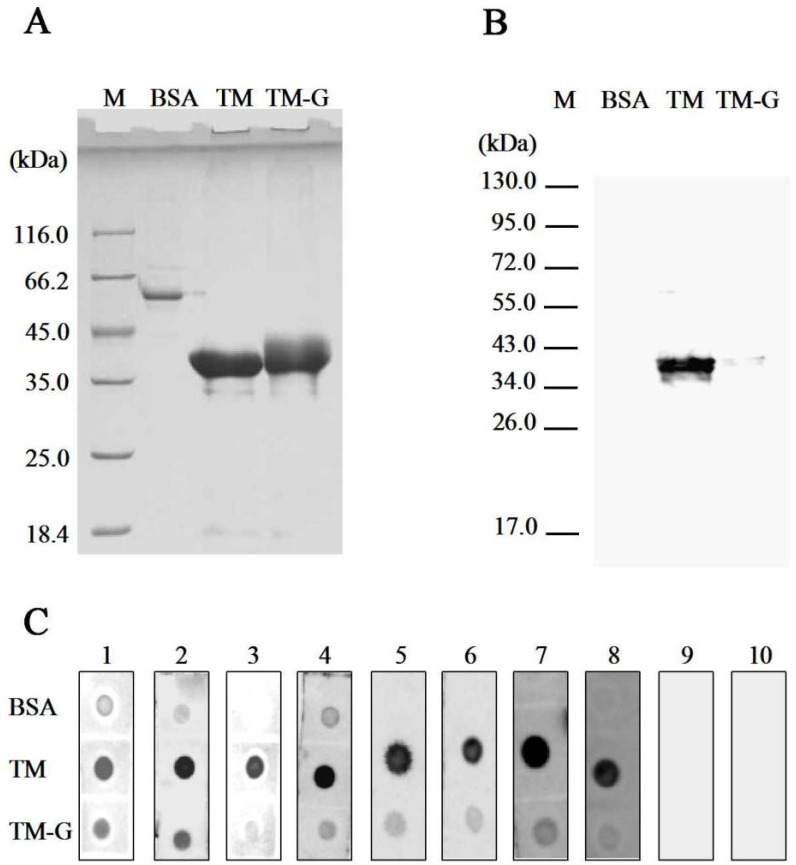
Immunoactivity analysis of TM-G: (**A**) SDS-PAGE analysis of TM-G. Lane M: protein marker. (**B**) IgG-binding capacity analysis of TM-G. Primary antibody: crab-TM-polyclonal antibodies, dilution 1:1 × 10^4^. Lane M: protein marker. (**C**) IgE-binding capacity analysis of TM-G. BSA was used as the negative control, TM was used as the positive control, and TM-G was used to compare with BSA and TM. Primary antibody: crab-sensitized patients’ sera (No. 1–8), negative sera (No. 9–10), dilution 1:5.

**Figure 2 molecules-27-02027-f002:**
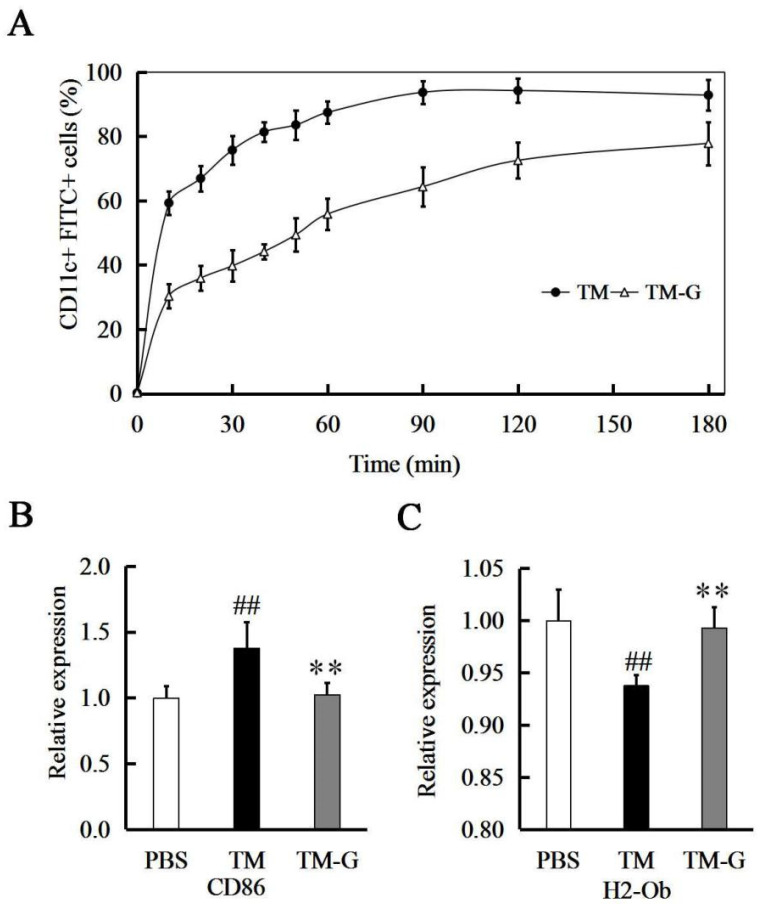
Allergen uptake and presentation by BMDCs: (**A**) TM-G uptake by BMDCs with different times (the gradient times were 0, 10, 20, 30, 40, 50, 60, 90, 120, and 180 min). (**B**,**C**) TM and TM-G presentation by BMDCs. All data are presented as the mean ± SD (*n* = 3). ^##^
*p* < 0.01 was used to compare with PBS group. ** *p* < 0.01 was used to compare with TM group.

**Figure 3 molecules-27-02027-f003:**
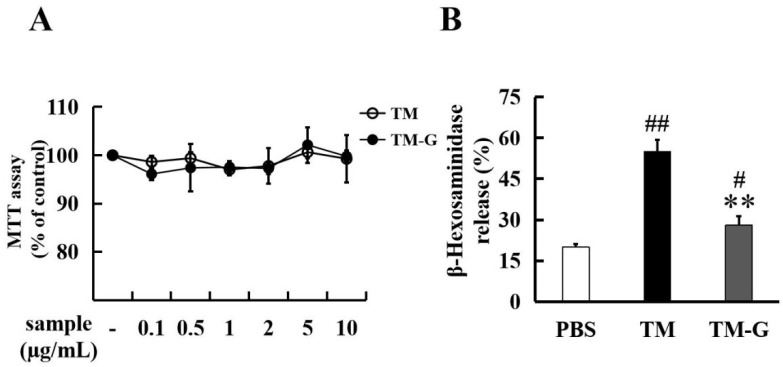
Bioavailability analysis of TM-G using RBL-2H3: (**A**) Cell viability analysis of TM and TM-G with different concentrations (0, 0.1, 0.5, 1, 2, 5, 10 μg/mL). (**B**) Effects of TM and TM-G on the release of β-hexosaminidase. All data are presented as the mean ± SD (*n* = 3). # *p* < 0.05 and ## *p* < 0.01 was used to compare with PBS group. ** *p* < 0.01 was used to compare with TM group.

**Figure 4 molecules-27-02027-f004:**
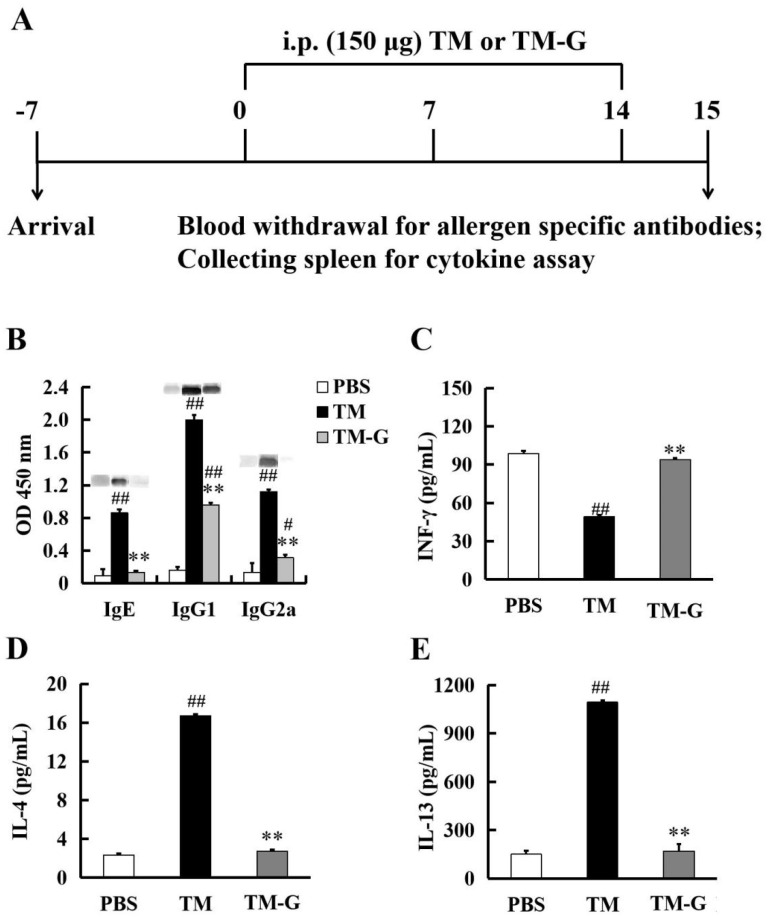
Sensitizing potential of TM-G: (**A**) Experimental design of in vivo sensitization. (**B**) Serum levels of IgE, IgG1, and IgG2a. (**C**–**E**) IFN-γ, IL-4, and IL-13 production by individual spleen lymph cells, was evaluated by ELISA in culture supernatants. All data are presented as the mean ± SD (*n* = 3). # *p* < 0.05 and ^##^
*p* < 0.01 was used to compare with PBS group. ** *p* < 0.01 was used to compare with the TM group.

**Figure 5 molecules-27-02027-f005:**
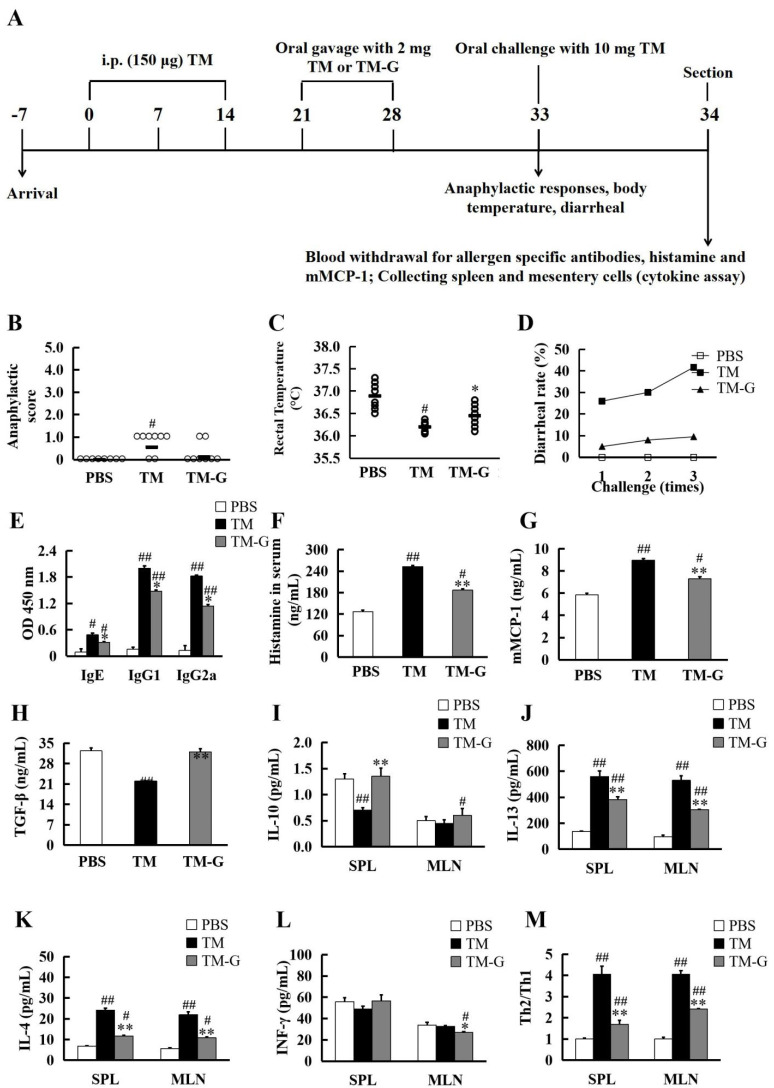
Oral tolerance induced by TM-G: (**A**) Experimental design of oral tolerance. (**B**) Anaphylactic scores. (**C**) Rectal temperature was measured 1 h after intragastric TM delivery. (**D**) Rates of diarrhea for 1 h after every challenge. (**E**) The levels of IgE, IgG1, and IgG2a in the serum of mice on day 34 were measured. (**F**–**H**) The levels of histamine, mMCP-1, TGF-β measured in sera (dilution 1:5, 1% skimmed milk). (**I**–**L**) The production of IL-10, IL-13, IL-4, and IFN-γ was measured in cultured spleen lymph cells and mesentery lymph cells, respectively. (**M**) The ratio of Th2/Th1. All data are presented as the mean ± SD (*n* = 3). ^#^
*p* < 0.05 and ^##^
*p* < 0.01 was used to compare with PBS group. * *p* < 0.05 and ** *p* < 0.01 was used to compare with TM group.

**Figure 6 molecules-27-02027-f006:**
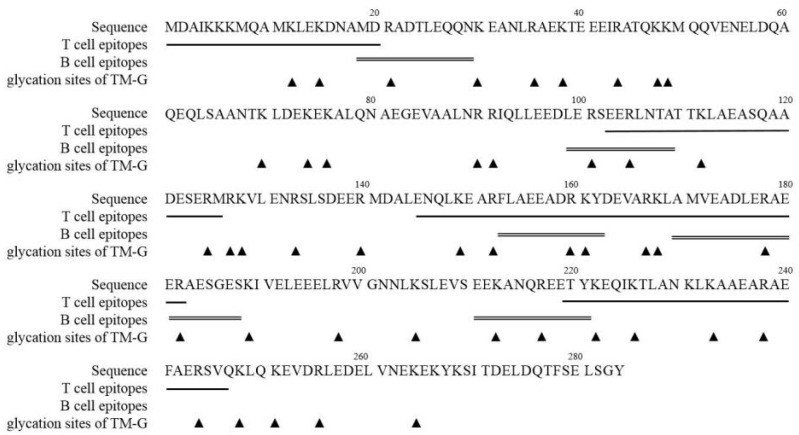
The modification analysis of the epitopes of TM-G. B cell epitopes of TM that have been identified, T cell epitopes of TM were predicted by prediction system. The T cell (underlined) and B cell (double underlined) reactive regions was compared with glycation sites of TM-G (marked by triangle).

## Data Availability

All data generated or analyzed during the present study are includedin this published article.

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
