# Peer review of "Reduction in Allergenicity and Induction of Oral Tolerance of Glycated Tropomyosin from Crab"

_molecules, 2022, doi:10.3390/molecules27062027_

Round 1
Reviewer 1 Report
The study, in a murine model, may have practical implications both at a clinical and industrial level, if supported by similar human data. The experimental model is fair and well structured and documented. There are imperfections. In particular, the reference citations are to be reviewed. It would be useful, for descriptive precision, to measure the IgE level for tropomyosin in the sera of patients sensitized to crab. In fact, the sensitization may be due to many other molecules such as sarcoplasmic calcium binding protein, argigine kinase, hemocyanin; myosin light chain, troponin C, myosin heavy chain, just to cite some of the molecules described in shellfish allergy.
...by immunoglobulin (Ig)E.1 ..... by immunoglobulin IgE [1]
The food allergens is abosorbed …. absorbed...
Author Response
A reply to the Reviewer 1
Comments and Suggestions for Authors
Q1: The study, in a murine model, may have practical implications both at a clinical and industrial level, if supported by similar human data. The experimental model is fair and well structured and documented. There are imperfections. In particular, the reference citations are to be reviewed. It would be useful, for descriptive precision, to measure the IgE level for tropomyosin in the sera of patients sensitized to crab. In fact, the sensitization may be due to many other molecules such as sarcoplasmic calcium binding protein, argigine kinase, hemocyanin; myosin light chain, troponin C, myosin heavy chain, just to cite some of the molecules described in shellfish allergy.
A1: Thanks for your revisions, and we apologize for any confusion. The references are revised one by one, and the details are as follows. In the introduction section (lines 30-88 in the revised manuscript), the numbers relating to the literature are modified to superscript. The publication time of reference 5 is corrected to 2008 (lines 407-408 in the revised manuscript). Meanwhile, The specific IgE levels with crab TM of positive sera were measured by ELISA previously. This segment has been added in line 116 in the revised manuscript and Table A. 1.
Q2: ...by immunoglobulin (Ig)E.1 ..... by immunoglobulin IgE [1]
The food allergens is abosorbed …. absorbed...
A2: Thank you very much for your careful reviewing of our manuscript. The words have been modified to “immunoglobulin E (IgE)” in line 31 and to “absorbed” in line 47 in the revised manuscript.

Reviewer 2 Report
Review molecules-1640301-peer-review-v1
General Comment
The manuscript of Xin-Yu Han1 , Tian-Liang Bai1 , Huang Yang1 , Yi-Chen Lin1 , Nai-Ru Ji1 , Yan-Bo Wang2 , Ling-Lin Fu2 , Min-Jie Cao1 , Jing-Wen Liu1 and Guang-Ming Liu1 *entitled: “Reduction in allergenicity and induction of oral tolerance of glycated tropomyosin from crab” submitted to Molecules journal, presents their work in the field of food allergy. MR usually occurs as a consequence of non-enzymatic glycosylation, and is now used as a potential tool to reduce the sensitivity of food allergens. Here, IgG TM-G binding activity was found to be reduced, and changes in the immune binding capacity of MRPs to murine specific IgE antibodies can be indirectly detected using the RBL-2H3 cell model. This study found that TM-G could inhibit the rate of β hexosaminidase release, suggesting that MRP immune binding activity IgE antibody from mouse serum was also significantly reduced. To analyze the immunogenic properties of TM-G, the effects on DC uptake activity by TM-G were examined by testing CD86 on DC and H2-Ob. In vivo experiments were also performed, where TM-specific IgE antibody in mouse serum was reduced in mouse-sensitive model, as far as possible showed the effect of MR on reducing allergen immunogenicity. The potential of TM-G to support the development of oral tolerance in mice were investigated.
Major Comments
- While the overall research design is appropriate, there is a room for improvement in terms of clarity of figures and data/results explanation. The best way to address this comment is to respond to all specific and minor comments outlined below.
- Style of the English language. If possible, engage English language editor with life science background to correct grammar and style. There are certain problems in English, in the sense that the order of words is not correct ... and that changes the concept of what you wanted to say. Is it tropomyosin from crab, or is it crab from tropomyosin?
- Abstract was written to me correctly, clearly, concisely with the idea of informing the reader what will be done in the mns.
- Introduction As for the introduction, it is too long, with sentences found in numerous papers regarding food allergies. A lot of good experiments were shown, but they are not cross linked properly, but I understand that this kind of writing is a challenge.
*Correctly cite the literature. Are the numbers relating to the literature large or small?
*Food allergy is an immune-mediated adverse reaction to food caused by allergens.2About 8% of infants and 2-3% of adults have a food allergy.3 Food allergens are usually some proteins or proteases with important structural or physiological functions in organisms. In my opinion, this sentence is absolutely unnecessary, I don't know if the journal prescribes the number of references ... but spending references to literally the same sentences that appear in the other pappers is meaningless.
*Shellfish is also considered as one of the most common aquatic products that can induce severe food allergy diseases; tropomyosin (TM) is the pan-allergen in shellfish.4 –not only in shellfish
*TM with the 31 to 42 kDa molecular weight (MW), is a type of acidic glycoprotein with good stability to protease, acid, and thermal treatmed.5-this is not correct reference
*The food allergens is absorbed by DCs and degraded into peptides by lysosomal after enter human body. The most resistant peptides are then being presented to T cell receptors through major histocompatibility complex (MHC) II molecules.12 The CD4+ helper T cells (Th) associated with the allergy cannot recognize allergen directly, only can recognize the MHC-II-antigenic peptide complex presented by dendritic cells.13 In addition, activated Th2 cells secrete interleukins 4 (IL-4) to regulate the differentiation of B cells into IgE secretory plasma cells and producing large amounts of IgE.14 Basophil degranulation has been used to study the allergic reactions which were induced by allergen cross-linking of IgE-bound to FcεR I receptors.15- I believe it is an awkward request, but I think there is a lot of description of the mechanism of antigen presentation, I think this paragraph should be rephrased and shortened it.
*In addition, Liu et al.22 explored that the crab of TM treated by enzyme cross-linking reaction has the potential to induce oral tolerance in mice, without explaining modifications in the structure of allergenic proteins. When you say this, it sounds like it's a crab from tropomyosin, not tropomyosin from crab.
*Antigenic epitopes are divided into two groups based on the cells which can be recognized: T cell epitopes and B cell epitopes.24 T cell epitopes are closely related to MHC, and allergic reactions are related to MHC-II.25 B cell epitopes can combine with specific IgE and IgG to stimulate effector cells degranulation and cause an allergic reaction.26- I would rephrase this, I think this is a science paper after all, and not a student book.
*Studies have shown that processed food could reduce allergen allergenicity by destroying antigenic epitopes.27- it can also lead to the creation of neoepitopes.
*In the present study, galactose could effectively decrease allergenicity in TM through MR in vitro assays.- Please make sure to correct in vitro throughout the paper, which should be in italics.
- Methodology
* Sera IgE antibodies (Table S. 1) to crab were measured by ImmunoCAP (Phadia AB, Uppsala, Sweden), when crab-specific IgE ≥ 0.35 kUA/L were defined as positive, negative sera IgE < 0.35 kUA/L, and all sera was stored at -80 ℃ until further use.- please match the tables in the paper and the supp info correctly.
*The hydrolysate was desalted and separated by capillary high-performance liquid chromatography and then analyzed by the mass spectrometer from Orbitrap-Fusion (Thermo Finnigan, San Jose, CA, USA). Finally, the sites where is modified by MR was compared with TM epitope.- please explain how mass spectrometry was done, what software was used, and how glycation sites were searched, or cite it.
- Results- In my opinion, the biggest complaint is the poor description of the results. Results are very interesting, but not explained. For example: The patients in the dot blot reacted very nicely, and nothing is described.
*Figure 1- I would discuss in one sentence the difference in staining between glycated and non-glycated in Figure 1A.
*TM-G uptake by BMDCs- Please provide label efficacy.
*Figure 2A- Also you can provide us a information, after how much time plato was reached.
*Figure 2(B,C) & 3(A,B)- please describe the picture nicely in captions.
* Table S2, S3- Please cite the tables from the supp info in the paper correctly.
- Dissccussion-General impression: more focused discussion, more references to support new ideas that are explained in the paper, whether a hypothesis is confirmed or not, but to refer as much as possible to tropomyosin.
*The MR is usually used in processed food commonly to improve food quality (citation) and taste, and now MR is gradually used to reduce the sensitivity of food allergen.-MR is a consequence of some method during food processing, and it is not the method itself.
*It has been previously shown the MRPs could promote the balance of Th2 cells and Th1 cells in mouse spleen lymphocytes, making them reach the cell level of normal mice.-citation
*Glycosylation sites determined by mass spectrometry have not even been discussed.
*The molecular mass of amino acid on the peptide segment could be analyzed using mass spectrometry.37- I'm not sure what this refers to and what the contribution of this sentence is.
* No row numbering, this makes review process more difficult.

Author Response
A reply to the Reviewer 2
Major Comments
Q1: While the overall research design is appropriate, there is a room for improvement in terms of clarity of figures and data/results explanation. The best way to address this comment is to respond to all specific and minor comments outlined below.
A1: Thank you for your suggestion. We have responded to all specific and minor comments outlined below and modified them according to your suggestion.
Q2: Style of the English language. If possible, engage English language editor with life science background to correct grammar and style. There are certain problems in English, in the sense that the order of words is not correct ... and that changes the concept of what you wanted to say. Is it tropomyosin from crab, or is it crab from tropomyosin?
A2: Thanks very much for your valuable reminder. We have invited native English speakers to revise the writing of the entire manuscript. We have modified and refactored according to your suggestions. We changed it to “... the TM of crab ....” in line 68, to “immunoglobulin E (IgE)” in line 31, to “absorbed” in line 47, to “absorption” in line 78, and to “different concentrations” in line 239 in the revised manuscript.,
Q3: Abstract was written to me correctly, clearly, concisely with the idea of informing the reader what will be done in the mns.
A3: Thanks very much for your valuable reminder. We modified abstract to “Tropomyosin (TM) is an important crustacean (Scylla paramamosain) allergen. This study aimed to assess Maillard-reacted TM (TM-G) induction of allergenic responses with cell and mouse models. We analyzed the difference of sensitization and the ability to induce immune tolerance between TM and TM-G by in vitro and in vivo models, then we compared the relationship between glycation sites of TM-G and epitopes of TM. In the in vitro assay, found out the sensitization of TM-G was lower than TM, and the ability to stimulate mast cell degranulation decreased from 55.07 ± 4.23% to 27.86 ± 3.21%. In the serum of sensitized Balb/c mice, the level of specific IgE produced by TM-G sensitized mice was significantly lower than TM, and the levels of interleukins 4 and interleukins 13 produced by Th2 cells in spleen lymphocytes decreased by 82.35 ± 5.88% and 83.64 ± 9.09%, respectively. In the oral tolerance model, the ratio of Th2/Th1 decreased from 4.05 ± 0.38 to 1.69 ± 0.19. Maillard reaction masked the B cell epitopes of TM and retained some T cell epitopes. Potentially, the MRPs as tolerance inducers are used for allergen-specific immunotherapy.” in lines 14-25 in the revised manuscript.
Q4: Introduction As for the introduction, it is too long, with sentences found in numerous papers regarding food allergies. A lot of good experiments were shown, but they are not cross linked properly, but I understand that this kind of writing is a challenge.
A4: Thank you for your suggestion. We have made the modification according to your suggestion. Firstly, we introduced food allergy and thermal treatment in lines 30-53 in the revised manuscript.
Secondly, we describe the current studies on reducing the allergenicity of food allergens by Maillard reaction, and introduced the oral tolerance mechanism of glycated food allergen in lines 54-77 in the revised manuscript.
Thirdly, the gaps in the three different locations have been integrated in lines 78-82 in the revised manuscript.
Finally, the purpose and significance of the research are briefly described in lines 83-88 in the revised manuscript.
The introduction was simplified, and the less relevant ones were deleted, which will be explained one by one later in the revised manuscript.
Q5: *Correctly cite the literature. Are the numbers relating to the literature large or small?
A5: Thanks for your revisions and we apologize for any confusion. In the introduction section (lines 30-88 in the revised manuscript) the numbers relating to the literature are modified to superscript, and other parts have also been checked.
Q6: *Food allergy is an immune-mediated adverse reaction to food caused by allergens.2About 8% of infants and 2-3% of adults have a food allergy.3 Food allergens are usually some proteins or proteases with important structural or physiological functions in organisms. In my opinion, this sentence is absolutely unnecessary, I don't know if the journal prescribes the number of references ... but spending references to literally the same sentences that appear in the other pappers is meaningless.
A6: Thank you very much for the advice. We modified it to “About 8% of infants and 2-3% of adults have a food allergy with increasing of incidence year after year.3 Food allergens are the key of food allergy, usually some proteins or proteases with important structural or physiological functions in organisms.” in lines 31-34 in the revised manuscript.
Q7: *Shellfish is also considered as one of the most common aquatic products that can induce severe food allergy diseases; tropomyosin (TM) is the pan-allergen in shellfish.4 –not only in shellfish
A7: Thanks very much for your kindly reminder. TM is the pan-allergen in shellfish and Arthropoda, such as Dermatophagoides pteronyssinus. For more accuracy, we modify it to “Shellfish is also considered as one of the most common aquatic products, that can induce severe food allergy diseases, tropomyosin (TM) is the pan-allergen.4” in lines 35-36 in the revised manuscript.
Q8: *TM with the 31 to 42 kDa molecular weight (MW), is a type of acidic glycoprotein with good stability to protease, acid, and thermal treatmed.5-this is not correct reference
A8: Thanks for your revisions and we apologize for any confusion. The publication time of reference 5 is corrected to 2008 (lines 407-408) in the revised manuscript, and other literatures are right.
Q9: *The food allergens is absorbed by DCs and degraded into peptides by lysosomal after enter human body. The most resistant peptides are then being presented to T cell receptors through major histocompatibility complex (MHC) II molecules.12 The CD4+ helper T cells (Th) associated with the allergy cannot recognize allergen directly, only can recognize the MHC-II-antigenic peptide complex presented by dendritic cells.13 In addition, activated Th2 cells secrete interleukins 4 (IL-4) to regulate the differentiation of B cells into IgE secretory plasma cells and producing large amounts of IgE.14 Basophil degranulation has been used to study the allergic reactions which were induced by allergen cross-linking of IgE-bound to FcεR I receptors.15- I believe it is an awkward request, but I think there is a lot of description of the mechanism of antigen presentation, I think this paragraph should be rephrased and shortened it.
A9: Thank you very much for the advice. We modified it to “The food allergens are absorbed by DCs and degraded into peptides by lysosomal after entering the human body. The peptide is recognized by CD4 + helper T cells (th) associated with allergy after binding to MHCII molecules.12, 13 In addition, activated Th2 cells secrete interleukins 4 (IL-4) to regulate B cells producing large amounts of IgE.14 Basophil degranulation has been used to study the allergic reactions with a glycated food allergen, and basophil degranulation was induced by allergen cross-linking of IgE-bound to FcεR I receptors.15” in lines 47-53 in the revised manuscript.
Q10: *In addition, Liu et al.22 explored that the crab of TM treated by enzyme cross-linking reaction has the potential to induce oral tolerance in mice, without explaining modifications in the structure of allergenic proteins. When you say this, it sounds like it's a crab from tropomyosin, not tropomyosin from crab.
A10: Thanks for your revisions and we apologize for any confusion. We changed it to “In addition, Liu et al.22 explored that the TM of crab treated by enzyme cross-linking reaction has the potential to induce oral tolerance in mice, without explaining modifications in the structure of allergenic proteins.” in lines 68-69 in the revised manuscript.
Q11: *Antigenic epitopes are divided into two groups based on the cells which can be recognized: T cell epitopes and B cell epitopes.24 T cell epitopes are closely related to MHC, and allergic reactions are related to MHC-II.25 B cell epitopes can combine with specific IgE and IgG to stimulate effector cells degranulation and cause an allergic reaction.26- I would rephrase this, I think this is a science paper after all, and not a student book.
A11: Thank you very much for the advice. We modified it to “Antigenic epitopes are divided into two groups based on the cells which can be recognized: T cell epitopes and B cell epitopes.24 Modifying B-cell epitope of allergen can reduce its allergic reaction, and retaining T-cell epitope of allergen can induce immune tolerance.25, 26” in lines 73-75 in the revised manuscript.
Q12: *Studies have shown that processed food could reduce allergen allergenicity by destroying antigenic epitopes.27- it can also lead to the creation of neoepitopes.
A12: Thanks very much for your kindly reminder. We changed it to “Studies have shown that processed food could change allergen allergenicity by destroying antigenic epitopes or creating neoepitopes.27” in lines 76-77 in the revised manuscript.
Q13: *In the present study, galactose could effectively decrease allergenicity in TM through MR in vitro assays.- Please make sure to correct in vitro throughout the paper, which should be in italics.
A13: Thank you very much for carefully reviewing of our manuscript, the word has been modified in italics in lines 84, 263, 338, and 453 in the revised manuscript.
Methodology
Q14: * Sera IgE antibodies (Table S. 1) to crab were measured by ImmunoCAP (Phadia AB, Uppsala, Sweden), when crab-specific IgE ≥ 0.35 kUA/L were defined as positive, negative sera IgE < 0.35 kUA/L, and all sera was stored at -80 ℃ until further use.- please match the tables in the paper and the supp info correctly.
A14: Thanks for your revisions and we apologize for any confusion. We changed it to “Table A. 1” in line 114 in the revised manuscript.
Q15: *The hydrolysate was desalted and separated by capillary high-performance liquid chromatography and then analyzed by the mass spectrometer from Orbitrap-Fusion (Thermo Finnigan, San Jose, CA, USA). Finally, the sites where is modified by MR was compared with TM epitope.- please explain how mass spectrometry was done, what software was used, and how glycation sites were searched, or cite it.
A15: Thank you very much for your question. Glycation sites testing was according to Han et al.28 with modifications. We change it to “The band of TM-G was hydrolyzed using trypsin (Promega, Madison, WI, USA) for 20 h, and then separated by capillary high-performance liquid chromatography (Orbitrap-ELite). The separated products were analyzed by Q-Exactive mass spectrometer. The result was named as “RAW file” and analyzed using the Maxquant software to find the modified amino acids.” in lines 179-183 in the revised manuscript.
Q16: Results- In my opinion, the biggest complaint is the poor description of the results. Results are very interesting, but not explained. For example: The patients in the dot blot reacted very nicely, and nothing is described.
A16: Thanks very much for your valuable reminder. We change it to “Moreover, the IgG/IgE-binding capacity of TM-G was lower than that of TM (Fig. 1B and C) using Western blot and Dot blot. The band of TM-G was not bound with the specific IgG antibody of TM. Compared with TM, TM-G exhibited weaker binding capacity with patients’ sera.” in lines 191-194 in the revised manuscript.
Q17: *Figure 1- I would discuss in one sentence the difference in staining between glycated and non-glycated in Figure 1A.
A17: Thanks very much for your valuable reminder. We added it to “The bands of TM and TM-G were represented in Fig. 1A. The MW of TM-G was higher than that of TM, and the band of TM-G performed diffusion.” in lines 190-191 in the revised manuscript.
Q18: *TM-G uptake by BMDCs- Please provide label efficacy.
A18: Thanks very much for your kindly reminder. We added it to “Uptake of TM and TM-G measured as the percent of CD11c+ and FITC-positive cells was time-dependent, while the mean fluorescence intensity continuously increased up to 180 min (Fig. 2A), the gradient time were 0, 10, 20, 30, 40, 50, 60, 90, 120, and 180 min. TM-G was taken up lower efficiently, reaching 72.53% of FITC-positive BMDCs, whereas BMDCs were incubated with FITC-labeled TM, the uptake ability was up to 94.23% (Fig. A. 1).” in lines 206-211 and 214-215 in the revised manuscript.
Q19: *Figure 2A- Also you can provide us a information, after how much time plato was reached.
A19: Thank you very much for your question and we are sorry for our improper description. Figure 2A is a line chart not a plato.
Q20: *Figure 2(B,C) & 3(A,B)- please describe the picture nicely in captions.
A20: Thank you for your suggestion. We have made the modification according to your suggestion.
Figure 2 (B, C) We changed it to “TM and TM-G presentation by BMDCs.” in captions in line 215 in the revised manuscript. And change the describe of result to “However, the expression level of CD86 on BMDCs stimulated by the TM-G group was significantly lower than that of the TM group (p < 0.01), without significant difference to PBS group (Fig. 2B). Furthermore, the expression level of H2-Ob was significantly up-regulated in the TM-G group in comparison to the TM group (p < 0.01), compared with the PBS group, the expression level of H2-Ob decreased by 6.24 ± 1.02% after stimulation in the TM group (Fig. 2C).” in lines 221-226 in the revised manuscript.
Figure 3 (A, B) We changed it to “(A) Cell viability analysis of TM and TM-G with different concentration (0, 0.1, 0.5, 1, 2, 5, 10 μg/mL). (B) Effects of TM and TM-G on the release of β-hexosaminidase.” in captions in lines 239-240 in the revised manuscript. And change the describe of result to “MTT assay was used to assess the cytotoxic effects of TM-G by RBL-2H3 cells. It was found that there was no effect on cell viability after TM and PBS stimulation. The cell viability rate of TM-G was more than 90% at different concentrations without significant difference to TM group (Fig. 3A). The sera of TM sensitized mice, using the 15-days injection model, was collected to sensitize RBL-2H3 cells. The effect of TM and TM-G on degranulation of RBL-2H3 cells was shown in Fig. 3B, compared with the PBS group, the degranulation rate of the TM group was significantly decreased (p < 0.01). The sensitization of TM-G was lower than TM, and the ability to stimulate mast cell degranulation decreased from 55.07 ± 4.23% to 27.86 ± 3.21%), meanwhile, the degranulation rate of the TM-G group was significantly increased compared with the PBS group (p < 0.05).” in lines 228-236 in the revised manuscript.
Q21: * Table S2, S3- Please cite the tables from the supp info in the paper correctly.
A21: Thanks for your revisions and we apologize for any confusion. We changed it to “Table A. 2 and Table A. 3” in lines 300-304 in the revised manuscript.
Q22: Dissccussion-General impression: more focused discussion, more references to support new ideas that are explained in the paper, whether a hypothesis is confirmed or not, but to refer as much as possible to tropomyosin.
A22: Thanks very much for your kindly reminder.
(1) We added reference 9 in lines 321-323 in the revised manuscript. We changed it to “In this study, the mechanism has been further clarified. We found that the IgG/IgE-binding activity of TM-G was reduced, Nakamura et al.9 found that the allergenicity of squid TM after MR was reduced, which is similar to this study.”
(2) We added new reference 33 in lines 325-328 in the revised manuscript. We changed it to “TM-G could significantly inhibit the β-hexosaminidase release rate, a similar found that after Maillard reaction, the degranulation of KU812 cells also decreased33, which indicated that the immune binding activity of MRPs to IgE antibody from mouse and human serum were also significantly decreased.”
(3) We added the reference 28 to compare of glycosylated residues in TM-G and TM with Maillard reaction with arabinose to discuss the glycosylation sites determined by mass spectrometry, it is “We found that TM-G has more glycosylated residues than TM with Maillard reaction with arabinose.28” in lines 353-354 in the revised manuscript.
Q23: *The MR is usually used in processed food commonly to improve food quality (citation) and taste, and now MR is gradually used to reduce the sensitivity of food allergen.-MR is a consequence of some method during food processing, and it is not the method itself.
A23: Thank you for your suggestion. We have made the modification according to your suggestion. We change it to “The MR usually occurred in processed food commonly to improve food quality and taste...” in line 318 in the revised manuscript.
Q24: *It has been previously shown the MRPs could promote the balance of Th2 cells and Th1 cells in mouse spleen lymphocytes, making them reach the cell level of normal mice.-citation
A24: Thanks very much for your kindly reminder. We added reference 28 in line 345 in the revised manuscript.
Q25: *Glycosylation sites determined by mass spectrometry have not even been discussed.
A25: Thank you for your suggestion. We found that TM-G has more glycosylated residues than TM with Maillard reaction with arabinose.28 in lines 353-354 in the revised manuscript.
Q26: *The molecular mass of amino acid on the peptide segment could be analyzed using mass spectrometry.37- I'm not sure what this refers to and what the contribution of this sentence is.
A26: Thank you for your suggestion. We have made the modification according to your suggestion. We added it to “Meanwhile, we explained the molecular mechanism of inducing tolerance of TM-G. Through the molecular weight of amino acids on the peptide can be analyzed by mass spectrometry38, the amino acid changes in TM-G were got. We found that TM-G has more glycosylated residues than TM with Maillard reaction with arabinose,28 and many glycosylated residues are located at T cell epitope or reported B cell epitope. Thus, the reaction of T cells could be activated without reacting with IgE on activating effector cells, so as to reduce allergic symptoms and improve the tolerance of the body.” in lines 351-357 in the revised manuscript.
Q27: *No row numbering, this makes review process more difficult.
A27: Thank you for your advice and we are sorry for our improper description. We added continuous row numbering in the revised manuscript.
